# Conservation laws in a neural network architecture: Enforcing the atom balance of a Julia-based photochemical model (v0.2.0)

Patrick Obin Sturm[1] and Anthony S. Wexler[1,2]

[1]Air Quality Research Center, University of California, Davis, California 95616 USA

[2]Departments of Mechanical and Aerospace Engineering, Civil and Environmental Engineering, and Land, Air and Water Resources, University of California, Davis, California 95616 USA

*Correspondence to*: P. Obin Sturm (posturm@ucdavis.edu) and Anthony S. Wexler (aswexler@ucdavis.edu)

**Abstract.** Models of atmospheric phenomena provide insight into climate, air quality, and meteorology, and provide a mechanism for understanding the effect of future emissions scenarios. To accurately represent atmospheric phenomena,
these models consume vast quantities of computational resources. Machine learning (ML) techniques such as neural networks have the potential to emulate compute-intensive components of these models to reduce their computational burden. However, such ML surrogate models may lead to nonphysical predictions that are difficult to uncover. Here we present a neural network architecture that enforces conservation laws to numerical precision. Instead of simply predicting properties of interest, a physically interpretable hidden layer within the network predicts fluxes between properties which are subsequently
related to the properties of interest. This approach is readily generalizable to physical processes where flux continuity is an essential governing equation. As an example application, we demonstrate our approach on a neural network surrogate model of photochemistry, trained to emulate a reference model that simulates formation and reaction of ozone. We design a physics-constrained neural network surrogate model of photochemistry using this approach and find that it conserves atoms as they flow between molecules, while outperforming two other neural network architectures in terms of accuracy, physical
consistency, and non-negativity of concentrations.

## 1 Introduction

One approach for increasing the computational efficiency of air quality and climate models is to replace the physical and chemical representation of atmospheric processes with machine learning surrogate models. Machine learning approaches for
surrogate models of phenomena in the atmospheric sciences emerged in the 1990s (Gardner and Dorling, 1998; Potukuchi and Wexler, 1997). However, these surrogate models might not necessarily (1) be faster than the reference model (Keller and Evans, 2019), (2) behave in a numerically stable way (Kelp et al., 2018; Brenowitz and Bretherton, 2018), or (3) make physical sense, for example by respecting deterministic constraints such as conservation laws (Keller and Evans, 2019; Kashinath et al., 2021). Recent efforts have taken steps towards the first two points, notably Kelp et al. (2020), who
demonstrate stability in recurrent, long-term predictions of gas-phase chemistry with a recurrent neural network architecture.

Their recurrent neural network is orders of magnitude faster than the reference model MOSAIC/CBM-Z (Zaveri et al., 2008).

Point 3 is an active area of research and the focus of this work. Complex machine learning (ML) tools, including neural
networks, can be criticized as being "black box" methods that have opaque inner workings: this criticism motivates the development of interpretable ML methods, or in the physical sciences, more *physically* interpretable ML (McGovern et al., 2019). Physics-informed neural networks exploiting automatic differentiation can reproduce numerical solutions to partial differential equations (Raissi et al., 2019). In the atmospheric sciences, physical information has been incorporated into machine learning models via balancing approaches after prediction (Krasnopolsky et al., 2010), a cost function penalizing
nonphysical behavior (Beucler et al., 2021; Zhao et al., 2019), including additional physically relevant information as input (Silva et al., 2021b), or incorporating hard constraints on a subset of the output in the neural network architecture (Beucler et al., 2019; Beucler et al., 2021).

Incorporating fundamental knowledge into ML algorithms will ensure adherence to the physical and chemical laws
underpinning these representations and likely improve the accuracy and stability of these algorithms. This work introduces a method to incorporate fundamental scientific laws in neural network surrogate models, in a way that ensures conservation of important quantities (for example mass, atoms, or energy) by imposing flux continuity constraints within the neural network architecture. Atom conservation is fundamental to atmospheric photochemistry and photochemistry is a computationally intensive component of these models so this work employs as an example inherently conserving atoms in a neural network
model of atmospheric photochemistry.

Recent efforts in machine learning methods for atmospheric chemistry have indicated physically informed ML as a future research direction (Keller and Evans, 2019; Kelp et al., 2020). Kelp et. al (2020) motivate exploring ML architectures that are customized with information about the systems they aim to model, and the potential for this to improve predictions of the
large concentration changes that frequently occur at the start of atmospheric chemistry simulations. Keller and Evans (2019) point out that incorporating physical information in ML, such as conservation laws, can help ensure point 2, numerical stability of ML, by keeping predictions within the solution space of the reference model. Keller and Evans (2019) also provide the example of atom conservation and propose inclusion of stoichiometric information as a possible solution and a future direction to explore. In this work, we focus on this latter goal: conserving atoms, much in line with the suggestions
outlined in Keller and Evans (2019), as well as the framework introduced by our prior work (Sturm and Wexler, 2020). More specifically, we utilize the weight matrix multiplication structure of a neural network (NN) to incorporate stoichiometric information in its architecture. The architecture of this physics-constrained model ensures conservation of atoms by including a constraint layer that has non-optimizable weights representing the stoichiometry of the reactions. The physics-constrained NN is trained to emulate a reference photochemical model simulating production and loss of ozone with 11

species and 10 reactions. A secondary benefit of the physics-constrained NN architecture is increased physical interpretability of the neural network: the output of the hidden layer before these constraints can be interpreted as the net flux of atoms between molecules, or in terms of chemical kinetics, the extent of reaction.

## 2 Derivation and model configuration

### 2.1 Physical constraints in the neural network architecture

Our prior work (Sturm and Wexler, 2020) introduced a framework that could be used with any machine learning algorithm

to introduce conservation laws. In the case of atmospheric chemistry, most ML surrogate model approaches have estimated future concentrations $C(t + \Delta t)$ from current concentrations $C(t)$ and other parameters $M(t)$, which can include meteorological conditions such as zenith angle, temperature, and humidity.

$$C(t + \Delta t) = F_C(C(t), M(t)) \; , \tag{1}$$

Rather than estimate the future value for the concentration (or more generally, the property of interest), we proposed training a machine learning algorithm to estimate fluxes *between* the properties of interest: for the photochemistry example, this is atom fluxes between molecules in a stoichiometrically balanced way. These fluxes are also interpretable as rates of reaction, or when integrated over a certain timestep, extents of reaction. The fluxes are related to the tendencies, or change of concentrations of species $\Delta C$, in a way that is stoichiometrically balanced. The stoichiometric information is contained in a

matrix $\mathbf{A}$ that relates fluxes, $S$, to change in concentrations, such that $\Delta C = \mathbf{A}S$. This framework leads to prediction of these fluxes using an ML algorithm that emulates

$$S(t + \Delta t) = F_S(C(t), M(t)) \, , \tag{2}$$

wherein $S$ is a vector of the time-integrated flux of atoms between model species due to photochemistry. Future concentrations can then be calculated via

$$C(t + \Delta t) = C(t) + \mathbf{A}S, \tag{3}$$

Typically, the reference model is used to generate training and test data sets to be used to develop the ML algorithm. However, $S$ values are not always standard output of such models: in some cases, the reference model can be altered to calculate and output $S$ values for the ML algorithm, for example calculating subgrid fluxes to train a physically consistent NN in a climate model (Yuval et al., 2021) or using explicit Euler integration in a simplified photochemical mechanism

(Sturm and Wexler, 2020). Unfortunately $S$ values often cannot be readily gleaned from the reference model for training a machine learning tool, especially when more sophisticated integrators are used. Our prior work focused on a way to invert $\mathbf{A}$ in order to calculate the target values $S$ (Sturm and Wexler, 2020). For the example of a surrogate model of condensation/evaporation in a sectional aerosol model, $\mathbf{A}$ is overdetermined and a left pseudoinverse exists (see Appendix A1). However, where there are more reactions than species of interest, such as in a photochemical system, or more generally when there are many different phenomena contributing to fewer quantities of interest, $\mathbf{A}$ will be underdetermined. This looks like $\mathbf{A} \in \mathbb{R}^{m,n}$ where $m < n$. For underdetermined systems, we applied a generalized inverse, restricted to lie in the space of all possible $S \in \mathbb{R}^n$, that would calculate $S$ from $\Delta C \in \mathbb{R}^m$. This approach does not guarantee that $S$ values would be realistic: sometimes predicted extents of reaction were erroneously negative in a photochemistry application.

This work explores the effects of implementing the $\Delta C = \mathbf{A}S$ step directly in the last layer of neural network as shown by Figure 1. Each node in a layer has an inner product between its weight and input vectors, $w^T x$. For the penultimate layer, the weight vector $w^T$ of each node corresponds to rows of $\mathbf{A}$. This can be thought of as the "constraint layer". The constraint layer has a zero bias vector and the linear activation function $f(x) = x$ such that the layer is simply a matrix operation equivalent to the $\mathbf{A}S$ product in equation 3. With this architecture, the inputs to this constraint layer are the time-integrated fluxes $S$ providing insight into the inner workings of the network as a side benefit. The activation function of the layer before should be chosen based on application. A rectified linear unit application that only outputs non-negative terms is appropriate for a photochemistry application, where integrated fluxes only have positive sign.

Including the $\mathbf{A}$ matrix representing the chemical system in the last layer of a neural network captures the coupling and interdependence of the different chemical species with custom, non-optimizable weights. Our approach resembles the Beucler et al. (2021) approach in that hard constraints are built into a neural network, with several key differences.

1. Our entire output vector represents a coupled system where all elements are subject to the constraints. This differs from the approach in Beucler et al. (2021). which constrains a chosen subset of the output and allows some output to be unconstrained.
2. This approach maintains our flux continuity constraint embodied in equation 3 (Sturm and Wexler, 2020).
3. Our approach does not require relating elements in the input to the output. Instead, the fluxes in the penultimate layer are related to the output such that tendencies are balanced.

Training the NN with $\mathbf{A}$ built into the last layer ultimately skips the compute-intensive and input-sensitive strategy of calculating the restricted inverse when the $\mathbf{A}$ matrix is underdetermined or rank deficient (Sturm and Wexler, 2020). This results in a neural network that conserves atoms in every prediction while also predicting the fluxes in the penultimate layer. This architecture adds physical interpretability to the last hidden layer of the neural network.

**2.2 Additional input to the neural network**

Physical information can be given as input to machine learning tools to improve predictions, for example when estimating aerosol activation fraction (Silva et al., 2021b). For our application, the complexity of the chemical system arises from the coupling of species, which interact with each other through chemical reactions. Bimolecular reactions (or generally reactions that involve two species) are often represented with rate laws of the form

$r = kC_iC_j$                                                                  (4)

where $r$ is the reaction rate for compounds $C_i$ and $C_j$ (the case $i = j$ is allowed) and $k$ is an often empirically determined reaction rate constant. In addition to the concentrations themselves, $C_iC_j$ can be calculated from the input concentrations and given as additional input to the neural network. Inclusion of this additional input, along with the methods described in

section 2.1, lead to our physics-constrained neural network model shown in Figure 1. We avoid use of physically-informed input to describe this approach, to prevent confusion with the physics-informed NN approach as introduced by Raissi et al. (2019). We do not call this approach "reactivity-informed" input, to keep the generalizability of this approach in mind. This additional input is informed by knowledge of bimolecular rate reactions: however, for other applications, additional input can take other forms. For the example of evaporation or condensation, the driving force of a concentration gradient could be

supplied as additional input.

What follows is an assessment of the accuracy of the physics-constrained neural network compared to a neural network with a "naïve" structure: neither a constraint layer nor additional input layer. To assess the relative contributions of each knowledge-guided adjustment to the neural network, we also construct an intermediate neural network, which contains the

additional knowledge-guided input but not the hard constraints built into the penultimate layer. Each network is trained to emulate the behavior of a reference model of chemistry modeling ozone production with 11 species and 10 reactions. All three are feedforward neural networks implemented in Python with the Keras library (Chollet et al., 2015) using a TensorFlow backend (Abadi et al., 2015).

Though the physics-constrained NN technically has two hidden layers, incorporating the flux-based balance in the second

hidden layer as a set of fixed weights with zero biases and linear activation functions adds no trainable parameters. This means that the penultimate layer is mapped to the output by a purely linear matrix operation. All three networks thus have only one hidden layer where parameters are adjusted during training. The width of this trainable hidden layer was chosen to contain 40 nodes. Each NN predicts an 11-element target vector of concentration tendencies $\Delta \boldsymbol{C}$. The naive NN takes a 13-element input vector, 11 concentrations as well as 2 additional inputs $\boldsymbol{M}$ based on meteorological conditions (sun angle).

The additional input of 5 bimolecular reactions to the intermediate and physics-constrained NNs results in 18-element input vectors. Increasing the size of the input layer adds 200 trainable parameters, so the intermediate and physics-constrained NN are significantly larger than the naïve NN. The intermediate and physics-constrained NN are comparable in terms of parameter space, with 1,211 and 1,170 trainable parameters respectively.


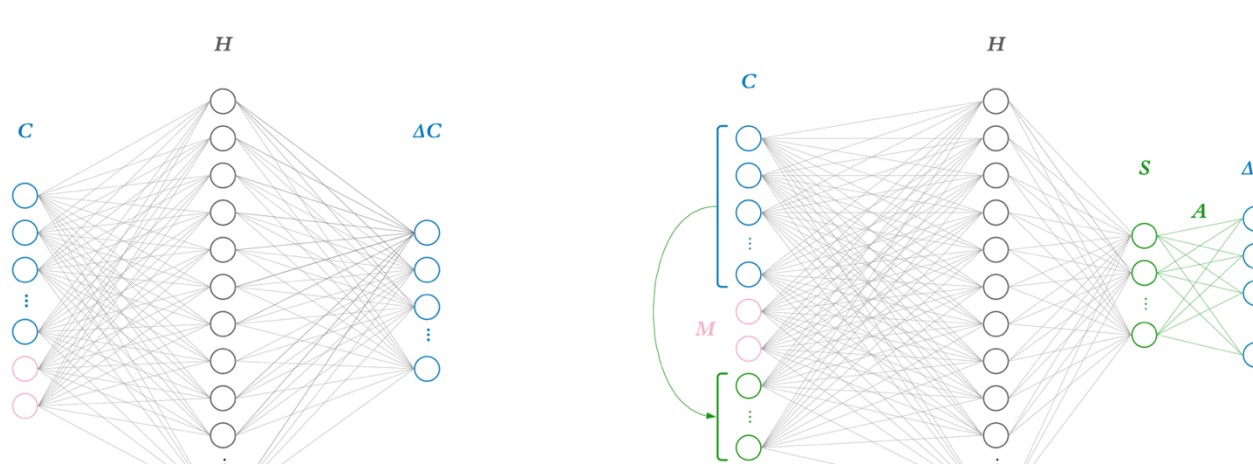

**Figure 1: Two neural network model architectures. The naïve neural network (left) takes as input concentrations for each of the eleven species $C$, as well as additional parameters M: in the photochemical surrogate model, M is the cosine of the zenith angle and change in cosine of the zenith angle. This input layer is fed to a hidden layer H, comprised of 40 nodes, each with weights, biases, and a rectified linear unit (ReLU) activation function. This is fed to a final output layer with a linear activation function and target values $\Delta C$ for the 11 species tracked in the reference photochemical model. The physics-constrained neural network (right)**
**includes additional input: 5 products of concentrations which resemble the rate law form of 5 bimolecular reactions. This is fed to a hidden layer the same size as that of the naïve NN: 40 nodes. The hidden layer is then fed to another layer $S$, which is chosen to have as many nodes as reactions in the chemical system (10), and ReLU activation functions to enforce non-negative output. This is subsequently fed via non-optimizable weights A and a linear activation function to the output vector $\Delta C$. The intermediate neural network (not pictured) contained the additional input informed by the rate laws, but is otherwise identical to the naïve**
**neural network.**

## 2.3 Reference photochemical model

To demonstrate the methods developed above, we used a simplified model for production of ozone, used by Dr. Michael Kleeman at the University of California, Davis for the course ECI 241 Air Quality Modeling. Our reference model focuses solely on gas-phase chemistry, including 10 reactions and 11 species. Table 1 includes the full list of reactions. Table 2 includes the full list of species, including whether they are active (reactants that influence reaction rates), steady-state, or build-up species. This simplified model still represents important features of ozone photochemical production, including $NO_X$ chemistry, VOC chemistry, peroxy radical, and hydroxyl radical. The limitations of this simple model mean that the subsequent neural networks will at best maintain these limitations. However, this simple example demonstrates conservation properties readily generalizable to larger, more sophisticated models of gas phase chemistry, such as CBM-Z (Zaveri and Peters, 1999), CBM-IV (Gery et al., 1989) and SAPRC (Carter, 1990; Carter and Heo, 2013).

We ported the reference model over from Fortran to Julia and adapted the model for use in this work, include varying cosine of zenith angle and other parameters, discussed further in section 2.4. The source code is available on Zenodo: https://doi.org/10.5281/zenodo.5736487. Julia was designed for its flexibility and ease of use, which is comparable to dynamic programming languages like Python, while allowing for computational performance approaching that of compiled languages like C or Fortran (Julia Documentation: https://julia-doc.readthedocs.io/en/latest/manual/introduction). These aspects of the Julia programming language have also motivated the recent development of JlBox, an atmospheric 0D box model written fully in Julia with gas-phase chemistry and aerosol microphysics (Huang and Topping, 2021).

| Table 1: Reactions | |
| --- | --- |
| Reaction | Reaction Number |
| $NO_2 + hv \rightarrow NO + O$ | R1 |
| $O + O_2 \rightarrow O_3$ | R2 |
| $O_3 + NO \rightarrow NO_2 + O2$ | R3 |
| $HCHO + hv \rightarrow 2\ HO_2^\cdot + CO$ | R4 |
| $HCHO + hv \rightarrow H2 + CO$ | R5 |
| $HCHO + HO^\cdot \rightarrow HO_2^\cdot + CO + H_2O$ | R6 |
| $HO_2^\cdot + NO \rightarrow OH^\cdot + NO_2$ | R7 |
| $OH^\cdot + NO_2 \rightarrow HNO_3$ | R8 |
| $HO_2H + hv \rightarrow 2\ OH^\cdot$ | R9 |
| $HO_2H + OH^\cdot \rightarrow H_2O + HO_2^\cdot$ | R10 |

| Table 2. Species | | |
|---|---|---|
| Name | Symbol | Species role |
| Ozone | $O_3$ | Active |
| Nitric oxide | NO | Active |
| Nitrogen dioxide | $NO_2$ | Active |
| Formaldehyde | HCHO | Active |
| Hydroperoxyl radical | $HO_2$· | Active |
| Hydrogen peroxide | $HO_2H$ | Active |
| Hydroxyl radical | OH· | Pseudo steady-state |
| Atomic oxygen | O | Pseudo steady-state |
| Nitric acid | $HNO_3$ | Build-up |
| Carbon monoxide | CO | Build-up |
| Hydrogen | $H_2$ | Build-up |


To fully represent the atom balance, the multitarget vector of tendencies $\Delta \boldsymbol{C}$ for both the naïve NN and physics constrained NN includes species that are not defined as "active species" in the reference model, including quickly reacting species that are modeled as pseudo-steady state and species that are only produced, called build-up species. These are summarized in

Table 2. Active species are defined in the original reference model as species that contribute to reaction rates, but have nonzero net rates of formation. Both NNs as depicted by Fig. 1 take concentrations of all 11 species as inputs, as well as cosine of zenith angle and change in cosine of zenith angle. The physics-constrained NN additionally takes 5 products of concentrations corresponding to the bimolecular reactions: R3 ($C_{O_3}C_{NO}$), R6 ($C_{HCHO}C_{OH}$), R7 ($C_{HO_2}C_{NO}$), R8 ($C_{NO_2}C_{OH}$), and R10 ($C_{HO_2H}C_{OH}$). Though R2 is also a bimolecular reaction, concentration of diatomic oxygen is assumed constant in the

reference model at a mixing 209,000 ppm, so the concentration product of the two reactants in R2 is proportional to the concentration of atomic oxygen. Assumption of diatomic oxygen as constant and a pseudo-infinite source/sink make it a special case: for this reason, oxygen is not included in Table 2 or in the stoichiometric balance in the following **A** matrix.

In both neural networks, the input layer is fed to a hidden layer of 40 nodes. While the naïve NN feeds this hidden layer to

the output vector, the physics-constrained NN contains a subsequent layer of 10 nodes corresponding to the fluxes of the 10 reactions: this penultimate layer is then connected to the output layer with non-optimizable weights corresponding to the **A**

matrix, to properly emulate the system of reactions. Within the framework outlined in section 2.1 and in Sturm and Wexler (2020), this system of reactions can be modeled by an 11 by 10 **A** matrix:

$$
\mathbf{A} =
\begin{bmatrix}
 & R1 & R2 & R3 & R4 & R5 & R6 & R7 & R8 & R9 & R10 \\
O_3 & 0 & 1 & -1 & 0 & 0 & 0 & 0 & 0 & 0 & 0 \\
NO & 1 & 0 & -1 & 0 & 0 & 0 & -1 & 0 & 0 & 0 \\
NO_2 & -1 & 0 & 1 & 0 & 0 & 0 & 1 & -1 & 0 & 0 \\
HCHO & 0 & 0 & 0 & -1 & -1 & -1 & 0 & 0 & 0 & 0 \\
HO_2 & 0 & 0 & 0 & 2 & 0 & 1 & -1 & 0 & 0 & 1 \\
HO_2H & 0 & 0 & 0 & 0 & 0 & 0 & 0 & 0 & -1 & -1 \\
OH & 0 & 0 & 0 & 0 & 0 & -1 & 1 & -1 & 2 & -1 \\
O & 1 & -1 & 0 & 0 & 0 & 0 & 0 & 0 & 0 & 0 \\
HNO_3 & 0 & 0 & 0 & 0 & 0 & 0 & 0 & 1 & 0 & 0 \\
CO & 0 & 0 & 0 & 1 & 1 & 1 & 0 & 0 & 0 & 0 \\
H_2 & 0 & 0 & 0 & 0 & 1 & 0 & 0 & 0 & 0 & 0 \\
\end{bmatrix}
$$

This rectangular matrix is rank deficient and obtaining extents of reactions $S$ from **A** and $\Delta C$ is a nontrivial inverse problem (Sturm and Wexler, 2020). **A** matrices of larger models, such as the version of CBM-Z implemented in the box model version of MOSAIC (Zaveri et al., 2008), are also rank deficient. Our simplified reference model shares this property with more sophisticated models, making it a good contender for a proof-of-concept implementation within a neural network.

Within modeling of chemical mechanisms, **A** is sometimes called the stoichiometry matrix. However, it can also be interpreted as the weighted, directed incidence matrix of the species-reaction graph of the chemical system. The species-reaction graph is a type of directed bipartite network that can give insight into a chemical system (Silva et al, 2021a). The species-reaction graph has two distinct sets of vertices corresponding to the reactions in Table 1 and species in Table 2: these vertices are connected by directed edges, corresponding to the values in the **A** matrix. Edges leaving a species vertex and going to a reaction vertex show that the species is a reactant and correspond to negative values in the **A** matrix. Similarly, edges leaving a reaction vertex and going to a species vertex show that the species is produced by that reaction: these edges correspond to positive values in the **A** matrix.

One metric of bipartite networks is the number of edges leaving nodes, called out-degree centrality. The out-degree centrality of a species vertex represents how many reactions the reactant participates in, and its value is the opposite sign of row sums of negative entries in the **A** matrix. The two species vertices with the highest out degree, 3, are formaldehyde (the sole reactive organic compound) and hydroxyl radical. Silva et al. (2021a) found that hydroxyl radical had the highest out-degree centrality in species-reaction graphs of 3 other chemical mechanisms. As in the other mechanisms, the out-degree centrality for the reactive nitrogen species, NO and NO₂, is higher than for other species. This indicates that, though simple,

the reference model is a relevant case study and the methods developed in this work show potential to be extended to other more sophisticated models of atmospheric chemistry.

**2.4 Training, validation, and test data**

Often, box model chemistry is an operator within a larger 3D transport model, which includes other operators modeling processes such as advection, emissions, and deposition. A good surrogate model should be able to emulate the input-output relationship of the reference model. If the context of machine learning surrogate modeling is operator replacement in larger chemical transport models (CTMs) or earth system models (ESMs), accurate short-term predictions on the order of the

operator splitting timestep are required. This context informs the strategy of emulating short-term behavior. We set up the reference model to write concentrations of the species every 6 minutes and train the neural network surrogate models to predict $\Delta C$ after this timestep. The timestep of 6 minutes is on the order of a common operator splitting timestep in a 3D chemical transport model: for example, the sectional aerosol model MOSAIC has a default timestep of 5 minutes (Zaveri et al., 2008). The operator splitting timestep in the 3D chemical transport model LOTOS-EUROS is chosen dynamically based

on wind conditions to satisfy the Courant-Friedrichs-Lewy criterion, but ranges between 1 and 10 minutes (Manders et al, 2017).

We used the reference model to generate 5,000 independent days of output, with concentrations of the 11 species reported every hour: 0.12 million 11-dimensional samples. For each day, concentrations were randomly initialized for active species,

documented in Table C1. The reference model was also adjusted to vary sunlight intensity, as measured by cosine of the zenith angle multiplied by a random factor associated with a full day simulation. This variable, as well as its change from the previous timestep, were chosen to be the additional parameters $M$ supplied to the neural network.

Of the 5,000 days, 4,800 were selected to be used as training and validation data for optimizing the neural network weights.

A portion of this data (10%) was designated as validation data: rather than optimizing the neural network parameters on this data, the model was evaluated on the validation data during training, with early stopping if no improvement was measured on this set. As in previous work (Kelp et al., 2020) we remove all samples from the training and validation data where ozone concentration exceeds 200 ppb. We additionally remove all days from the test data where ozone exceeds 200 ppb at any point, resulting in 126 full days used to evaluate the accuracy of the neural networks.


For supervised machine learning, the inputs $X$ ($C$ and $M$ concatenated, as well as concentration products for the second neural network) are different from the targets $\Delta C$. With a similar transformation, the inputs can be normalized on a scale from 0 to 1. This can be done by scaling each input feature $x$ in $X$ by its corresponding maximum and minimum in the training data:

$$x = \frac{x - x_{min}}{x_{max} - x_{min}} \tag{5}$$

This information can be put into a diagonal matrix $\mathbf{N_{X,maxmin}}$ whose elements are $x_{max} - x_{min}$ for each input. Representing the input minimums for each element as a vector $X_{min}$, the normalized feature space in this case looks like

$$X_{norm} = N_{X,maxmin}^{-1}(X - X_{min}) \tag{6}$$

This is implemented in Python using the sci-kit learn preprocessing tool MinMaxScaler (Pedregosa et al., 2011).

## 3 Results

### 3.1 Comparing neural networks with and without physical constraints

Figure 2 shows predictions of $\Delta C$ compared to the reference model for the first four species by the naïve NN (orange, top row), the intermediate NN (purple, middle row) and the physics-constrained NN (green, bottom row). These tendencies were more accurately predicted when incorporating physical information into the neural network, showing $R^2$ values of 0.95 or higher when evaluated on the test data.

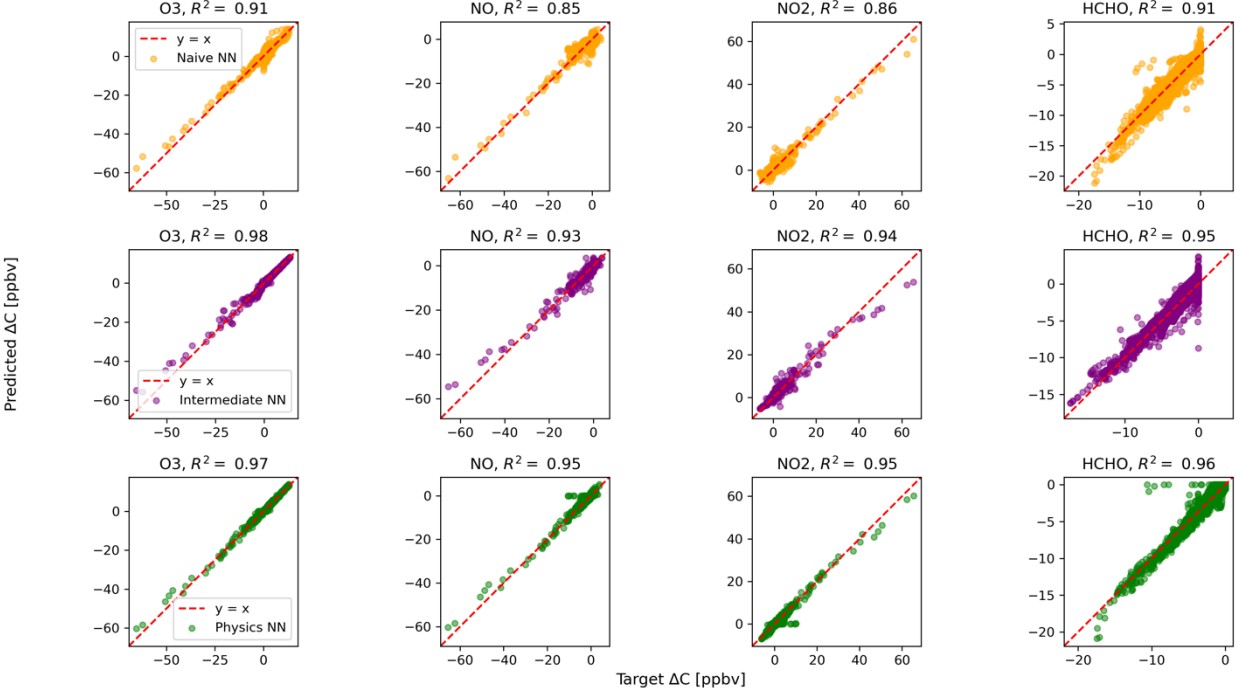

**Figure 2: Scatter plots of target values to predicted values, for naïve NN (orange, top) and the physics-constrained NN (green, bottom), on 126 test days, data the NNs did not see during training. The diagonal dashed line in red is the 1:1 line.**

The scatter plots of tendencies for the other active species and the buildup species are shown in Fig. B1 and Fig. B2 in Appendix C. Both the naïve and physics-constrained architectures show poor accuracy (negative $R^2$ values) in predictions

of $\Delta C$ for hydrogen peroxide. This can be attributed partially to the tendency range for hydrogen peroxide, which is 2 orders of magnitude smaller than that for other compounds. Error for species with smaller changes in concentration might be improved with choice of a different loss function than mean squared error (MSE) between predictions and targets, but a normalized MSE loss function heavily biased towards zero values for the tendency vector $\Delta C$ led the neural network to only predict zero values for all species: this approach was ruled out early on. The intermediate NN has an $R^2$ value of 0.82 for

hydrogen peroxide, but also overestimates the magnitudes of hydrogen peroxide tendency (in both negative and positive directions). The physics-constrained NN demonstrates improved predictions of tendency for hydroperoxyl radical, with an $R^2$ value of 0.86, better than those of the naïve NN (0.42) and intermediate NN (0.65).

Of the buildup species, only hydrogen predictions do not improve with the intermediate and physics-constrained

architectures: $R^2$ values for the naïve, intermediate, and physics-constrained NNs are comparable at 0.87, 0.85, and 0.86, respectively. The other two build-up species, nitric acid and carbon monoxide are better predicted by the other two NNs, as shown in Figure B2. Nitric acid and carbon monoxide are two compounds necessary for the conservation of nitrogen and carbon in the overall system.

The naïve NN demonstrated $\Delta C$ predictions outside of the solution space of the reference model. Though more accurate than the naïve NN, the intermediate NN also demonstrates $\Delta C$ predictions that are outside of the solution space of the reference model. Figure 2 shows that some of the predictions for formaldehyde tendency using the naïve and intermediate NNs are positive, despite there being no source for formaldehyde in the reference model: all reactions including formaldehyde are sinks, where it either reacts with another species or undergoes photolysis. The physics-constrained NN restricts all

predictions of formaldehyde tendency to be at most zero, which is in line with it being only a reactant. Similarly, some naïve and intermediate NN predictions of $\Delta C$ for the build-up species (species that are only products) are negative: this can be seen in Figure B2 in Appendix B. The reference model has no sinks for these build-up species, which are strictly products of reactions, hence the term "build-up". The physics-constrained NN restricts the elements $\Delta C$ corresponding to build-up species to a positive half-space.


Evaluated on overall metrics using a test data set, the physics-constrained NN outperforms both other NN architectures. The maximum absolute errors of $\Delta C$ in the test data set for the naïve, intermediate, and physics-constrained NNs are 10.7 ppb, 11.6 ppb and 10.6 ppb, respectively. For all species, the naïve NN predicted $\Delta C$ within 1.31 ppb, the intermediate NN predicted $\Delta C$ within 0.84 ppb, and the physics-constrained NN predicted $\Delta C$ within 0.76 ppb for 99% of cases. Error metrics

evaluated with all active and build-up species for the 126 independent test days are given in Table 3. The physics-

constrained NN is more accurate than both other NN architectures in these overall metrics, despite having a slightly smaller trainable parameter space than the intermediate NN.

| Table 3. Comparison of NN models | | | | | |
|---|---|---|---|---|---|
| Neural Network | Mean absolute error [ppb] | RMSE [ppb] | Normalized mean absolute error | Maximum absolute error [ppb] | 99[th] percentile absolute error [ppb] |
| Naïve NN | 0.18 | 0.33 | 0.868 | 10.7 | 1.32 |
| Intermediate NN | 0.11 | 0.22 | 0.547 | 11.6 | 0.84 |
| Physics NN | 0.07 | 0.20 | 0.337 | 10.6 | 0.76 |

### 3.2 Performance over varying concentration scales

At the beginning of simulations, steep changes in concentration occur when the chemical system is initialized in a state far from pseudo equilibrium. Chemical operators within larger models approach this pseudo equilibrium, but in each operator splitting time step other operators such as advection and emission perturb these concentrations back away from the pseudo equilibrium state. This informs the focus on short-term accuracy, and the target vector of tendencies $\Delta C$ after timesteps of 6 minutes.

Kelp et al. (2020) found that the most long-term stable models came at the price of diminished accuracy in predictions of the extreme $\Delta C$ at the beginning of simulations, and motivated further research of ML models with specialized architectures. Our approach is only used for short term predictions. However, the two architectures with additional input informed by the reactivity show an ability to predict the $\Delta C$ at the beginning of each full-day simulation, while also remaining accurate relative to the naïve neural network in conditions that have smaller changes in concentration; those that occur after the initial transient return to the pseudo equilibrium condition.

The large $\Delta C$ values resulting from randomly initialized states far from equilibrium are well modeled by both the intermediate NN and the physics-constrained NN: this can be seen, for example, by ozone in Figure 2. Figure 3 shows scatter plots of $\Delta C$ as predicted by the 3 NN models, when only evaluated on 23-hour runs after the first hour of simulation that includes the transient return to pseudo equilibrium. Figure 3 and its reported $R^2$ metrics are analogous to Figure 2, with the difference being that Figure 3 repeats the analysis with the first hour of each day removed from the test data. With the first hour removed, the changes in ozone concentration shrink by a factor of ~4. While the naïve NN shows a substantial drop in accuracy of $\Delta C$ for reactive nitrogen species, the physics-constrained NN shows a smaller change in its $R^2$ metric.

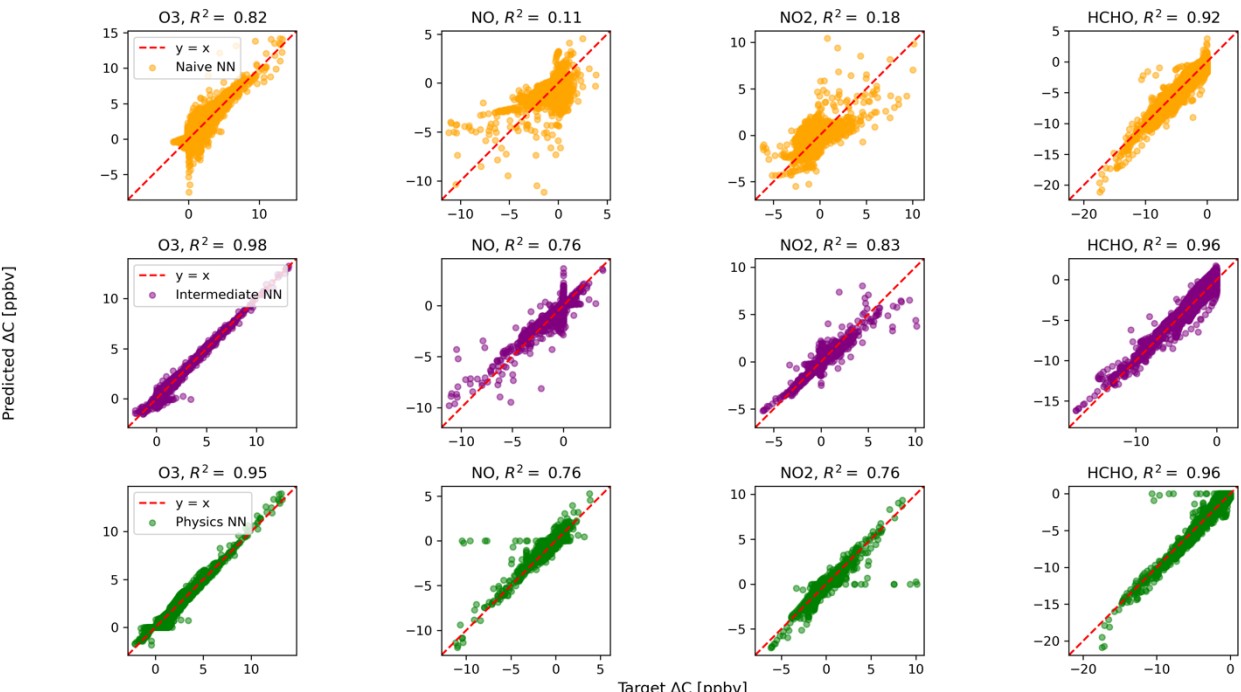

**Figure 3: Scatter plots of target values to predicted values, for naïve NN (orange, top) and the physics-constrained NN (green, bottom), on 126 test days, for 23 hour runs excluding the first hour of simulation.**

Silva et al. (2021b) found that a physically regularized NN emulator of aerosol activation fraction outperformed its naïve counterpart, especially in the edge case of prediction values falling within the lower 10% of the possible range. Similarly, we find that our NNs that have additional chemically relevant input (the intermediate and physics-constrained NNs) are much more accurate for $\Delta \boldsymbol{C}$ of NO and $NO_2$ than the naïve NN when only evaluated on cases falling within the lower ~25% of the range of test data: Fig. 3 illustrates this improvement. Though both the intermediate and physics-constrained NNs better predict tendencies of the species than the naïve NN, this improvement is magnified when disregarding large concentration changes that occur at the beginning of simulations.

### 3.3 Steady-state species

Under the pseudo-steady state assumption that certain species have near-zero net rates of change in concentration, their concentrations become algebraic expressions of the concentrations of other species in the system. In our reference model, these steady state species are hydroxyl radical OH and atomic oxygen O. Approximating the rates of change for each steady state species to be zero and isolating their concentrations as expressions of other concentrations and rate constants, we obtain the following equations for $C_{OH}$ and $C_O$:

$$C_{OH} = \frac{k_7 C_{HO2} C_{NO} + 2k_9 C_{HO_2H} * hv}{(k_6 C_{HCHO} + k_8 C_{NO_2} + k_{10} C_{HO_2H})}$$ (10)


and

$$C_O = \frac{k_1 C_{NO2} * hv}{(k_2 C_{O_2})}$$ (11)

where the $k_j$ correspond to rate constants for reactions j = 1, 2,…, 10, and $C_{O_2}$ is assumed constant at $2.09 \times 10^5$ ppm. Information on $hv$ is included in $M(t)$.

The physics-constrained NN predicts $\Delta C$, which is added to $C(t)$ to calculate $C(t + \Delta t)$. Then the steady-state concentrations at time $t + \Delta t$ are determined by the concentrations of active species using Eqs. (10) and (11). Figure 4 shows

the scatter plots of concentrations using predictions from the physics-constrained NN versus the reference model.

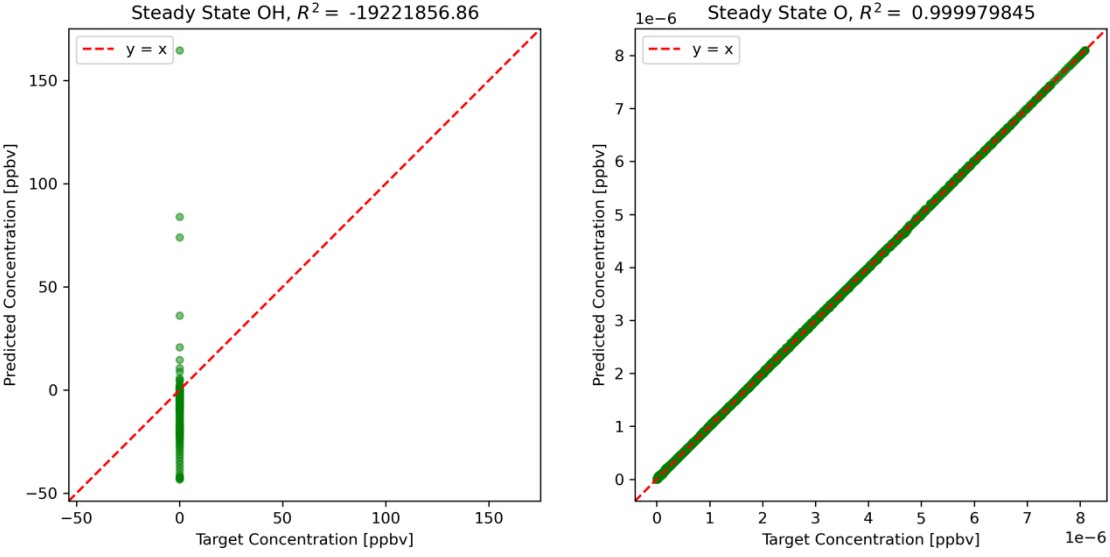

**Figure 4: Predictions of steady-state species using NN predictions for all other species concentrations.**

The concentration of atomic oxygen is a function of only one variable influenced by NN predictions, $NO_2$ concentration, and

is nearly perfectly predicted. The concentration of the hydroxyl radical is dependent on concentrations of 5 other species, and is very sensitive to small errors in some of the species: $HO_2$, NO, and to some extent formaldehyde. The limitation of the physics-constrained NN to predict OH indicates a that additional physical information might need to be included in order to

optimize the physics-constrained NN to predict OH accurately, e.g. including Eq.(10) in the objective function when optimizing NN parameters.


### 3.4 Atom conservation in the physics-constrained neural network

The balance imposed on species by the physics-constrained neural network results in conservation of the total carbon and nitrogen. The atom balance for carbon and nitrogen can be demonstrated by summing up the mixing ratios of species these atoms occur in, multiplied by the number of atoms within that species. Figure 5 shows that there is a net zero change in total carbon and nitrogen in the system when using the physics-constrained NN. Balances of oxygen and hydrogen are not shown: oxygen is not conserved, because of the treatment within the reference model of diatomic oxygen as an infinite source and sink. Hydrogen is not conserved because $H_2O$ is not explicitly tracked.

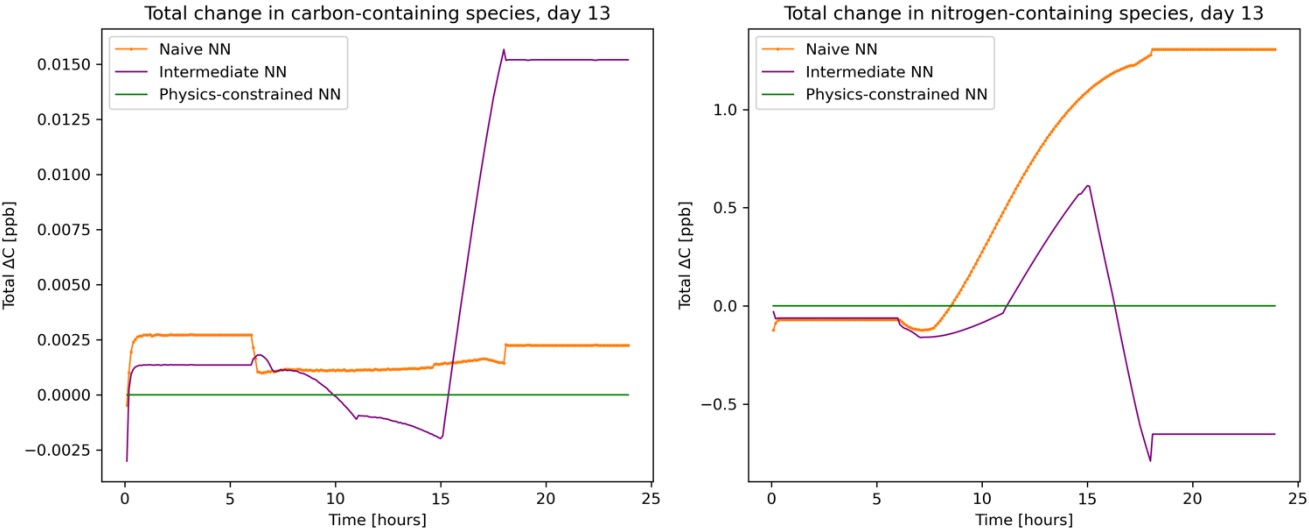


**Figure 5: Net tendency of carbon-containing species on the left and nitrogen-containing species on the right, as predicted by the naïve NN (orange, squiggly line with textured points), the intermediate NN (purple, squiggly line) and the physics-constrained NN (green, flat line). While the naïve neural network predictions lead to fluctuations in the overall carbon and nitrogen budget of the system, the physics-constrained neural network conserves the total amount of both.**


With every prediction, the naïve and intermediate NNs remove or add some carbon and nitrogen to the system. Though errors are small in the representative day shown in Figure 5, this error occurs every 6 minutes. Summed up over the day, the naïve NN predictions lead to a net addition of ~0.4 ppb of carbon-containing species and net addition of ~147 ppb of nitrogen-containing species; the intermediate NN predictions lead to a net addition of 1.2 ppb of carbon-containing species and a net removal of 39.5 ppb of nitrogen-containing species.


### 3.5 Preventing negative concentrations

Though the physics-constrained neural network inherently balances mass, there is no built-in constraint to ensure nonzero concentrations: predicted tendencies might exceed the magnitude of their corresponding concentrations in the previous timestep. However, the number of negative concentrations was reduced by a factor of more than 17 when using the physics-constrained NN compared to the naïve NN. The naïve NN predictions led to 44,017 negative concentrations and the physics-constrained NN predictions led to 2,489 negative concentrations in the test dataset containing 272,160 values (including active and build-up species but excluding pseudo steady-state species whose concentrations are not calculated by adding their corresponding tendencies). Put another way, predictions by the naïve NN led to 13.2% of the values becoming negative, with the most negative concentration at -7.5 ppb. Predictions by the intermediate NN led to 11.7% of the values becoming negative, with the most negative concentration at -4.4 ppb. In contract, predictions by the physics-constrained NN led to approximately 0.7% of concentrations becoming negative, with the most negative concentration at -3.7 ppb. Despite not being explicitly enforced, the physics-constrained NN still performed much better than both other NNs, predicting over an order of magnitude fewer nonphysical negative concentrations on the test dataset. Predictions from the naïve and intermediate NNs also led to negative values for some of the buildup compounds, which is outside of the solution space of the reference model: build-up species are initialized at zero concentration and are only products of reactions. The architecture of the physics-constrained NN enforces non-negativity of the penultimate layer corresponding to fluxes and positive coefficients in the stoichiometric weight matrix $A$ corresponding to build-up species: this ensures that all $\Delta C$ predictions for build-up species are positive and therefore all concentrations.

### 4 Conclusions

Machine learning algorithms have potential to efficiently emulate complex models of atmospheric processes, but purely data-driven methods may not respect important physical symmetries that are built into the classical models, such as conservation of mass or energy. Prior efforts (Sturm and Wexler, 2020) developed a framework for building in conservation laws to machine learning algorithms: by using the relationship between fluxes and tendencies in systems, the fluxes can be posed as learning targets for the ML algorithms, and then tendencies can be predicted in a balanced manner. This work builds on that framework and proposes implementing the flux-tendency relationship directly into the architecture of a neural network, so that the neural network will inherently respect the conservation laws, much like the reference model it emulates. As an example of how this framework can be implemented, we design a physics-constrained neural network surrogate model of photochemistry with input resembling bimolecular reaction rates, and a penultimate hidden layer enforcing an atom balance. The weights for the penultimate layer are hard stoichiometric constraints and can be obtained via the approach in Sturm and Wexler (2020) relating tendencies of molecular species to atom fluxes between them.

Adding additional parameters based on physical information (in this case chemical reaction rates) improve predictions, as demonstrated by the intermediate NN and the physics constrained NN. Like previous work (Silva et al, 2021b) these NNs more accurately predict edge cases than the naïve NN: in our case, lower $\Delta C$ conditions after the first hour of simulation approaching pseudo equilibrium. However, improved accuracy of the intermediate NN does not correspond to an adherence to physical laws. Both the naïve and intermediate NNs deliver solutions outside of the solution space of the reference model, including negative tendencies for purely buildup species and positive formation of species that were purely reactants. Their predictions also lead to high numbers of negative concentrations, which are nonphysical. Most importantly, material is not conserved by either the naïve NN or intermediate NN: only the physics-constrained obeyed the stoichiometric atom balance that is a fundamental property of chemical reactions. The results of this study show promise for hybrid models that combine our knowledge of physical processes with data-driven machine learning approaches, and motivates future exploration of other physically interpretable machine learning techniques that can incorporate additional prior information such as pseudo steady-state approximations.

The reference model used in this work shares important chemical properties with more sophisticated models, making this approach readily extendable to detailed chemical mechanisms. In extension to larger models, the effect of varying hyperparameters, including input and output dimensionality, network depth (number of layers) and layer width (number of nodes in the layers), will have to be assessed. Such a study may be better suited for application to a more sophisticated reference model that has a higher dimensionality and more realistic inputs, such as varying temperature. This approach also has potential to be integrated into work studying the speedup potential of neural networks versus their reference models, also better suited for studies of larger, more sophisticated and computationally intensive reference models. The primary purpose of this work is to illustrate the atom balance enforced by the architecture of the physics-constrained neural network. We observe a secondary effect: that by including built-in information about the chemical system, both the atom balance and the input proportional to the instantaneous reaction rates of the bimolecular reactions, accuracy of the neural network is improved.

**Appendix A: Calculating a left pseudoinverse for condensation/evaporation in a sectional aerosol model**

This framework has been demonstrated for photochemistry but can be generalized to other applications, such as change of concentrations of condensable species in a sectional aerosol model, for example MOSAIC (Zaveri et al., 2008). Recent work has been published on machine learning surrogate models for aerosol microphysics (Gettelman et al., 2021; Harder et al., 2021). Harder et al. (2021) have indicated mass conservation as a future research direction for ML surrogate models of aerosol microphysics, and have proposed regularization via a cost function or post-prediction mass fixers. Training ML algorithms on target fluxes $S$ rather than tendencies $\Delta C$ would allow for mass conservation to numerical precision if the tendencies are related to fluxes via an $\mathbf{A}$ matrix as in Sturm and Wexler (2020). Studying the system of equations modeling evaporation and condensation in a sectional model, we see that a left pseudoinverse of the corresponding $\mathbf{A}$ matrix can be used to obtain fluxes $S$ from concentrations (typical model output), unlike the rank-deficient $\mathbf{A}$ matrix in the photochemical application focused on in this work and Sturm and Wexler (2020).

Both mass transfer and thermodynamics play a role in the transport of material between the gas and aerosol phases (Wexler and Seinfeld, 1991). This idea can be represented by a system of equations taken directly from equations 3 and 4 in Zaveri et al. (2008), relating change in concentration to flux between the gas and particle phases:

$$\frac{dC_{a,i,m}}{dt} = k_{i,m}\left(C_{g,i} - C^{*}_{a,i,m}\right) \tag{A1}$$

$$\frac{dC_{g,i}}{dt} = -\sum_m k_{i,m}\left(C_{g,i} - C^{*}_{a,i,m}\right) \tag{A2}$$

where $C_{g,i}$ is the gas-phase bulk concentration of species $i$, $C_{a,i,m}$ is the aerosol-phase concentration of species $i$ in size bin $m$, $C^{*}_{a,i,m}$ is the partial pressure of species $i$ in bin $m$ in equilibrium with $C_{a,i,m}$, and $k_{i,m}$ is a first order mass transfer coefficient for species $i$ in bin $m$. This system of equations can be put in matrix form, with change in concentration as a vector on the left-hand side, and column coefficients multiplying the flux values $S_{i,m} = k_{i,m}\left(C_{g,i} - C^{*}_{a,i,m}\right)$ on the right-hand side of the equation.

Below is an illustrative example of what the $\mathbf{A}$ matrix for evaporation/condensation could look like for a single species ($i = \{1\}$) example with one gas phase and by 8 size bins, which is a standard bin amount used in WRF-Chem. The rates of change of concentration are related to their fluxes by a matrix $\mathbf{A_1}$,

$$\begin{pmatrix} d_t C_g \\ d_t C_{a,1,1} \\ d_t C_{a,1,2} \\ \vdots \\ d_t C_{a,1,8} \end{pmatrix} = \mathbf{A_1} \begin{pmatrix} S_{1,1} \\ S_{1,2} \\ \vdots \\ S_{1,8} \end{pmatrix} \tag{A3}$$

Where

$$\mathbf{A_1} = \begin{bmatrix} -1 & -1 & -1 & -1 & -1 & -1 & -1 & -1 \\ 1 & 0 & 0 & 0 & 0 & 0 & 0 & 0 \\ 0 & 1 & 0 & 0 & 0 & 0 & 0 & 0 \\ 0 & 0 & 1 & 0 & 0 & 0 & 0 & 0 \\ 0 & 0 & 0 & 1 & 0 & 0 & 0 & 0 \\ 0 & 0 & 0 & 0 & 1 & 0 & 0 & 0 \\ 0 & 0 & 0 & 0 & 0 & 1 & 0 & 0 \\ 0 & 0 & 0 & 0 & 0 & 0 & 1 & 0 \\ 0 & 0 & 0 & 0 & 0 & 0 & 0 & 1 \end{bmatrix} \tag{A4}$$

This overdetermined 9 by 8 matrix accounts for fluxes from the gas phase to each different bin. With more species, say, 23
species, the system resembles a block matrix:

$$\mathbf{A} = \begin{bmatrix} \mathbf{A_1} & \mathbf{0} & \cdots & \mathbf{0} \\ \mathbf{0} & \mathbf{A_2} & & \mathbf{0} \\ \vdots & & \ddots & \vdots \\ \mathbf{0} & \mathbf{0} & \cdots & \mathbf{A_{23}} \end{bmatrix} \tag{A5}$$

With the assumption that all species have the same number of bins, $\mathbf{A_1} = \mathbf{A_2} = \cdots = \mathbf{A_{23}}$. $\mathbf{A}$ is overdetermined and has full
column rank, meaning that there are more equations than unknowns. This makes calculating a unique left inverse possible:

$$\mathbf{A_L} = (\mathbf{A^T A})^{-1} \mathbf{A^T} \tag{A6}$$

The existence of a unique $\mathbf{A_L}$ is useful because concentration values (and therefore $\Delta C$) might be more easily obtainable from
reference models than the right-hand-side integrated flux values $S$. From $\Delta C$, $\mathbf{A_L}$ can be used to obtain $S$ values:

$$S = \mathbf{A_L} \Delta C \tag{A7}$$

which a supervised machine learning algorithm can be trained to predict from concentration, temperature, and other
parameters.

**Appendix B: Scatter plots of active and buildup species**

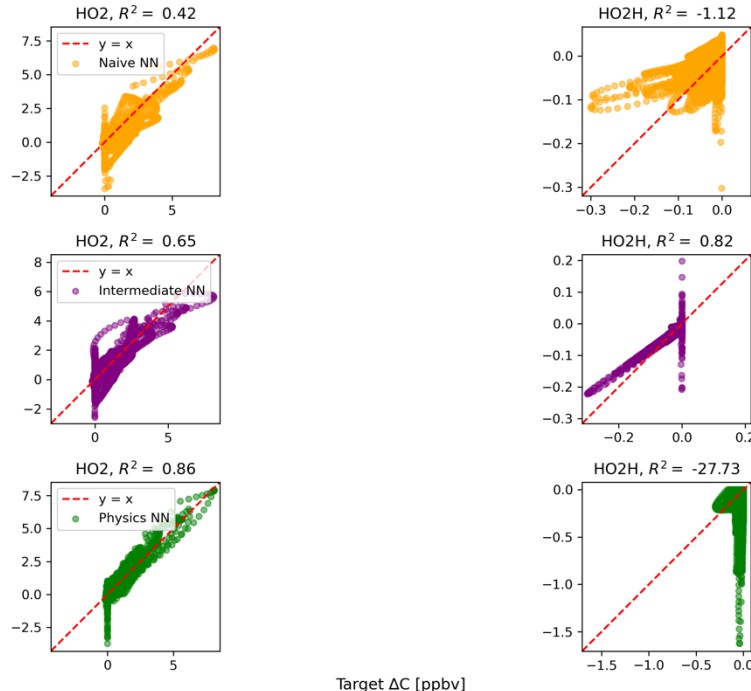

**Figure B1. Scatter plots of $\Delta C$ for active species 5 and 6, as predicted by the naïve NN and physics-constrained NN.**

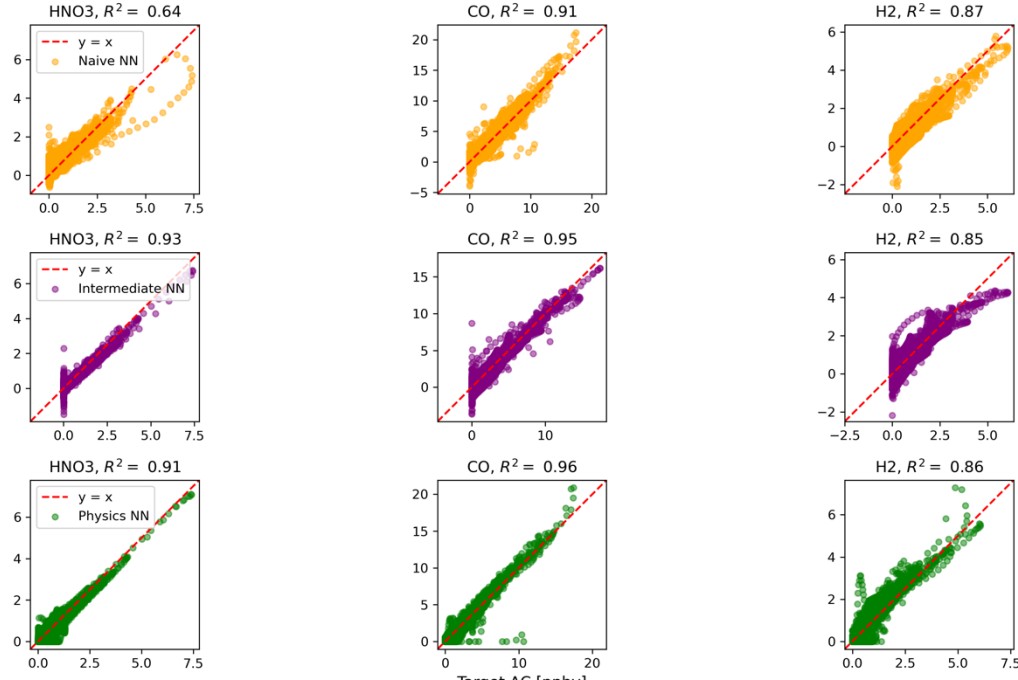

**Figure B2. Scatter plots of $\Delta C$ for buildup species as predicted by the naïve NN and physics constrained NN.**


## Appendix C: Reference model initialization

The 5000 independent days include randomly initialized values for active species concentrations at the beginning of each day of simulation. Cosine of zenith angle is also multiplied by a random factor between 0 and 1 for the day, to vary intensity of photolysis reactions. Steady-state concentrations are a direct function of active species concentrations, so are initialized accordingly. Build-up species concentrations are initialized at zero.

| Table C1. Initialization of active species concentrations | | | |
|---|---|---|---|
| Name | Symbol | Range | Distribution |
| Ozone | $O_3$ | 0.001 - 0.1 ppm | logarithmic |
| Nitric oxide | NO | 0.0015 - 0.15 ppm | logarithmic |
| Nitrogen dioxide | $NO_2$ | 0.0015 - 0.15 ppm | logarithmic |
| Formaldehyde | HCHO | 0.02 - 2 ppm | logarithmic |
| Hydroperoxyl radical | $HO_2\cdot$ | 0 - 0.00001 ppm | linear |
| Hydrogen peroxide | $HO_2H$ | 0 - 0.01 ppm | linear |


*Code and data availability*

The exact version of the Julia reference model used to generate model output for the neural networks is archived on Zenodo at https://doi.org/10.5281/zenodo.5736487. To maximize accessibility, the model output as text files is available for download without needing to run the reference model (as *S.txt*, *C.txt* and *J.txt*). These text files are used in a Python script with the exact version used to construct, train, and evaluate the neural networks available at https://doi.org/10.5281/zenodo.6363763. At this DOI, the neural networks are also available for download in hierarchical

data format (.h5).

*Author contributions*

ASW initiated the project and conceptualized the flux balancing framework and the additional input. POS created the concept of the physics-constrained NN architecture in this work, and developed the model code using Julia for the reference

model and Python for the neural networks. ASW and POS designed the experiments and wrote the manuscript together.

*Competing interests*

The authors declare that they have no conflict of interest.


*Acknowledgements*

This research is supported by the UC Davis CeDAR (Center for Data Science and Artificial Intelligence Research) Innovative Data Science Seed Funding Program. We would also like to acknowledge Michael J. Kleeman at UC Davis, who contributed the original reference model in Fortran that we later ported to Julia and adapted for our application.

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
