# Peer review of "Conservation laws in a neural network architecture: Enforcing the atom balance of a Julia-based photochemical model (v0.2.0)"

_Geoscientific Model Development, 2021_

## Author Comment (AC1)

**Author response to referee and community comments on Conservation laws in a neural network architecture: Enforcing the atom balance of a Juliabased photochemical model (v0.2.0) P. Obin Sturm and Anthony S. Wexler**

We would like to thank the referees for their time and detailed feedback. We appreciate the valuable comments that contribute to improving this work.

Below, we respond point-by-point to referee and community comments. Referee and community comments are in *italic*, our responses are in **bold**, and updates to the manuscript are shown in blue.

**Anonymous Referee #1**

The authors present a method to make a surrogate model of a photochemistry model of low to intermediate complexity, including an interpretable physical constraint in the neural network. The work is very relevant both from the mathematical (machine learning) concept and for the application in atmospheric modelling. Mathematical concepts are combined with physical/chemical interpretations and the demonstration model is well chosen to be relevant by itself and being complex enough to demonstrate the generalizability of the method to larger models. Methods and technical results are mostly presented in a concise and clear way.

I have some minor comments and suggestions to improve the readability of the paper. The method is demonstrated for a photochemical model which can be an aim by itself. This combination of the concept with a clear demonstration is on the one hand the strength of the paper. On the other hand, making clear reference to the demonstration case in the earlier (more conceptual) part of the paper can help the reader.

**Detailed comment:**

**Abstract**

The context can be narrowed on the one hand (atmospheric composition/air quality) and widened on the other (impact of emission changes AND climate change) on atmospheric composition/air quality. Would be good to more specifically mention what the demonstration model is. Photochemistry is important part of air quality models, can be aim by itself, and in addition this model contains all essential elements for generalization.

**Response:** We include a sentence emphasizing the generalizability of this approach, while more specifically describing the reference model. The abstract now reads:**

Models of atmospheric phenomena provide insight into climate, air quality, and meteorology, and provide a mechanism for understanding the effect of future emissions scenarios. To accurately represent atmospheric phenomena, these models consume vast quantities of computational resources. Machine learning (ML) techniques such as neural networks have the potential to emulate compute-intensive components of these models to reduce their computational burden. However, such ML surrogate models may lead to nonphysical predictions that are difficult to uncover. Here we present a neural network architecture that enforces conservation laws to numerical precision. Instead of simply predicting properties of interest, a physically interpretable hidden layer within the network predicts fluxes between properties which are subsequently related to the properties of interest. This approach is readily generalizable to physical processes where flux continuity is an essential governing equation. As an example application, we demonstrate our approach on a neural network surrogate model of photochemistry, trained to emulate a reference model that simulates formation and reaction of ozone. We design a physics-constrained neural network surrogate model of photochemistry using this approach and find that it conserves atoms as they flow between molecules, while outperforming two other neural network architectures in terms of accuracy, physical consistency, and non-negativity of concentrations.

**Introduction**

The authors could point out what is different with atmospheric reactions with resperct to other ML approaches with conservation laws at the beginning of the introduction. (1 55-60) would make a nice part of the introductoin.

**Response: We have made the first sentence from (l 55-60) the first sentence of the introduction. We have moved the rest from the beginning of section to the introduction, now (l 49-61), as well as an additional sentence. This section, now in the introduction, reads:**

Incorporating fundamental knowledge into ML algorithms will ensure adherence to the physical and chemical laws underpinning these representations and likely improve the accuracy and stability of these algorithms. This work introduces a method to incorporate fundamental scientific laws in neural network surrogate models, in a way that ensures conservation of important quantities (for example mass, atoms, or energy) by imposing flux continuity constraints within the neural network architecture. Atom conservation is fundamental to atmospheric photochemistry and photochemistry is a computationally intensive component of these models so this work employs as an example inherently conserving atoms in a neural network model of atmospheric photochemistry.

As a reference to the general context of machine learning in earth science, the paper by Kashinath gives a good perspective and deserves citation.

Kashinath et al 2021 Physics-informed machine learning: case studies for weather and climate modelling Phil. Trans. Royal Sociaety 20200093https://doi.org/10.1098/rsta.2020.0093

**Response:** We agree – goals and perspectives from that paper are much in line with our prior work from 2020, as well as the approach developed in this manuscript. This is now cited in the first paragraph.

L37: Reference to Beucler et al 2019 would also be relevant, where also the neural network architecture itself is used for constraint, not just part of the output cost function. Beucler T, Rasp S, Pritchard M, Gentine P. 2019 Achieving conservation of energy in neura network emulators for climate modeling. (http://arxiv.org/abs/1906.06622).

**Response:** We agree – we referenced this in our prior work (the cited companion paper, Sturm and Wexler 2020) and now include the reference to Beucler et al 2019 in this manuscript as well.

**2 Derivation and model configuration**

Figure 1 caption explicity mentions 11 species and additional parameters, but the photochemical model is not mentioned in the text yet. Would also good to mention the demonstration model very briefly in the introduction. Or leave out the explicit reference to the number 11 and the meteorological parameters.

**Response:** Thanks for pointing this out. We introduce this now with sufficient detail in the last few sentences of the introduction:

The architecture of this physics-constrained model ensures conservation of atoms by including a constraint layer that has non-optimizable weights representing the stoichiometry of the reactions. The physics-constrained NN is trained to emulate a reference photochemical model simulating production and loss of ozone with 11 species and 10 reactions. A secondary benefit of the physics-constrained NN is increased physical interpretability of the neural network: the output of the hidden layer before these constraints can be interpreted as the net flux of atoms between molecules, or in terms of chemical kinetics, the extent of reaction.

2.3 Would be better to first describe the gas-phase photochemistry model, then the Julia motivation.

**Response:** We agree. We moved the Julia motivation to the second paragraph of section 2.3, after describing the mechanism and referring to Tables 1 and 2.

O2 is missing from Table 2. Diatiomic oxygen is present at much higher concentrations that 0.209 ppm, makes nearly 21% of troposphere so I suspect at unit error (absolute mixing ratio). Cf. line 329.

**Response:** Thanks for catching the unit error. We do not include diatomic oxygen in the stoichiometric balance: we have adjusted the relevant sentences in lines 279-283 as follows:

Though R2 is also a bimolecular reaction, concentration of diatomic oxygen is assumed constant in the reference model at a mixing 209,000 ppm, so the concentration product of the two reactants in R2 is proportional to the concentration of atomic oxygen. Assumption of diatomic oxygen as constant and a pseudo-infinite source/sink make it a special case: for this reason, oxygen is not included in Table 2 or in the stoichiometric balance in the following **A** matrix. Figure 2 caption: Sentence can be misunderstood, what do you mean by 'data the NNs were not optimized to predict'

Response: We have changed this to read: data the NNs did not see during training

Around line 300, Figure 3, I'm confused with the 23 hours. Is Figure3 equal to Figure 2 but then withouth the first hour of simulation of each simulation day?

Response: Yes, that's correct. We have updated this paragraph with an additional sentence to make this clear. The sentence that was originally "For instance, when the first hour is removed from the test data, the changes in ozone concentration shrink by a factor of ~4" is now

Figure 3 and its reported  $R^2$  metrics are analogous to Figure 2, with the difference being that Figure 3 repeats the analysis with the first hour of each day removed from the test data. With the first hour removed, the changes in ozone concentration shrink by a factor of ~4.

*3.4: Figure 6 is not present and the text 1349 seems to refer to Figure 5.* **Response: Thank you, this is fixed in the updated manuscript.**

**Anonymous Referee #2**

The authors consider neural network surrogate models with physics constraints to enforce conservation laws, they demonstrate the effectiveness of their method using a photochemistry model. Experiment results show perfect atom conservation. The presentation is clear. It is an important research topic to enforce conservation laws (e.g., mass conservation) in neural network surrogates.

**Some comments on the paper:**

The authors should refer to and discuss the paper: Beucler et al. 2019 "Achieving Conservation of Energy in Neural Network Emulators for Climate Modeling" (arxiv.org/abs/1906.06622). In that paper, two options to implement physics constraints were discussed: 1. Constraining the loss function; 2. Constraining the architecture. The approach of this paper belongs to option 2 (figure 2, Beucler et al. 2019). It presents a detailed derivation and implementation of the constrained layer with fixed weights for a photochemical model.

Line 105, it is stated that a key difference with the work of Beucler et al. 2021 is "1. Our entire output is calculated under the constraints, rather than only a portion of the output." In my opinion, this statement is not correct. In the general architecture (Beucler et al.), parts of the output can be constrained and some parts of the output don't need to be constrained. It's a choice, to constrain the entire output is just a particular case.

Response: That is a good point. This statement should be rewritten to say that all of our output is constrained, rather than a chosen number of output elements subject to n constraints as per the approach in Beucler et al. 2021. Our third point was also unclear: unlike the Beucler et al., approach, the constraints do not relate input to the output. Below are the changes we have made accordingly.

Including the **A** matrix representing the chemical system in the last layer of a neural network captures the coupling and interdependence of the different chemical species with custom, non-optimizable weights. Our approach resembles the Beucler et al. (2021) approach in that hard constraints are built into a neural network, with several key differences.

- Our entire output vector represents a coupled system where all elements are subject to the constraints. This differs from the approach in Beucler et al. (2021). which constrains a chosen subset of the output and allows some output to be unconstrained.
- 2. This approach maintains our flux continuity constraint embodied in equation 3 (Sturm and Wexler, 2020).
- 3. Our approach does not require relating elements in the input to the output. Instead, the fluxes in the penultimate layer are related to the output such that tendencies are balanced.

The authors compared their constrained NN with a simple one-layer NN (naïve NN). The constrained NN has two layers (though the  $2^{nd}$  layer has fixed weights). A two-layers NN could (possibly) improve the performance and it is also more comparable to their constrained NN, it would be fair to compare the constrained NN with a 2-layers naïve NN.

**Response:**

Though the physics-constrained NN has two layers, the nature of the fixed weights in the second layer add no trainable parameters (and is a purely linear matrix operation). Without the inclusion of the additional input, the 2 layer physics-constrained NN actually has fewer trainable parameters than the naïve NN: 970 compared to 1,011. The second layer adds 121 non-trainable parameters.

However, The inclusion of the additional input adds 200 trainable parameters. As suggested by the community comment, we now include an intermediate NN in the study which uses the additional input without the constraints layer. This intermediate NN has 1,211 trainable parameters, more the physics-constrained NN, which has 1,170 trainable parameters. For this reason, the model comparison is now quite fair in terms of parameter space. We believe that by including the intermediate NN as suggested by the community comment, we address the reasoning behind this suggestion as well.

**Section 2.2**

Though the physics-constrained NN technically has two hidden layers, incorporating the fluxbased balance in the second hidden layer as fixed weights with zero biases and linear activation functions adds no trainable parameters. This means that the penultimate layer is mapped to the output by a purely linear matrix operation. All three networks thus have only one hidden layer where parameters are adjusted during training. The width of this layer was chosen to contain 40 nodes. Each NN predicts an 11-element target vector of concentration tendencies  $\Delta C$ . The naive NN takes a 13-element input vector, 11 concentrations as well as 2 additional inputs M based on meteorological conditions (sun angle). The additional input of 5 bimolecular reactions to the intermediate and physics-constrained NNs results in 18-element input vectors. Increasing the size of the input layer adds 200 trainable parameters, so the intermediate and physics-constrained NN are significantly larger than the naïve NN. The intermediate and physics-constrained NN are comparable in terms of parameter space, with 1,211 and 1,170 trainable parameters respectively. *Some minor textual comments:*

- (1), use a different symbol for the function C (C already refers to concentrations);
- (2), use a different symbol for function S (the  $2^{nd} S$ )
- Response: We now use the symbols  $F_C$  and  $F_S$  to represent the functions of C and S respectively.
- Line 221, 1.2 million samples should be: 0.12 million samples Response: We correct this in the updated manuscript.
- ٠
- Line 232, "126 days with continuous observations", data in each day are generated/initialized separately (see Line 220), so they are not continuous?
  Response: Good point for clarity, we changed this to 126 full days
- Line 240, Svector à S vector Response: We corrected this to simply say vector.

**Community Comment 'Model comparison and naming conventions', Oscar Jacquot**

Dear authors,

I would like to make a few comments comments about your paper :

1) This comment concerns the choice of your baseline model and subsequently the way some results regarding your approch are assessed.

In section 2 you introduce two networks. One is a simple two layers dense neural network which you refer to as naïve. The other network, which is the focus of your work, features two architecture modifications with regards to the naïve network. One affects the input with the addition of bilinear terms correponding to the two bodies reactions in your chemical system. Another modification affects the output, resulting in your network effectively predicting the rate of each reaction in the second to last layer and subsequently assessing the change in concentration of each species in the last layer, according to stochiometric constraints.

In section 3.4, you compare carbon and nitrogen conservation between the two networks, effectively showing that your stochiometric constraint is indeed verified while the naïve implementation fails to do so. This is indeed a predictable property from your network, as this constraint is in some sense hard coded and is independent from the ability of your network to predict reaction rates well.

Yet, one could reasonnably assume that providing the naïve network with relevant bilinear terms as an input without including hard coded stochiometric information would on its own improve forecast performance. Therefore, I believe that your paper currently has a blind spot : what are the relative contributions of the the input modification and the stochiometric constraint to the improvement of forecasting ability with regards to the naïve network ?

I think that your paper would benefit from displaying the results obtained with a third intermediate network, which would include the input modification but no stochiometric constraints on the output. The comparison of correlation and element conservation for all 3 networks would be more insightful, but should not require much work as the same learning procedure can be applied for the intermediate network.

Response: Thank you for this comment. We have decided to include an intermediate NN based on your suggestion and agree that this benefits the study, while also addressing a suggestion from a referee comment on how to make NNs more comparable in terms of trainable parameter space. This is a valuable addition that does not require major changes to the manuscript, but allows for a more thorough investigation of the difference between the naïve and physics-constrained NN, and the effect of the hard constraints via the A matrix. Below are added areas in the text.

**Section 2.2**

To assess the relative contributions of each knowledge-guided adjustment to the neural network, we also construct an intermediate neural network, which contains the additional knowledge-guided input but not the hard constraints built into the penultimate layer. Each network is trained to emulate the behavior of a reference model of chemistry modeling ozone production with 11 species and 10 reactions. All three are feedforward neural networks implemented in Python with the Keras library (Chollet et al., 2015) using a TensorFlow backend (Abadi et al., 2015).

**Caption of Figure 1: We add**

The intermediate neural network (not pictured) contained the additional input informed by the rate laws, but is otherwise identical to the naïve neural network.

For Section 3 (Results), we change Figures 2, 3, and 5 to include the intermediate NN, as well as add a row to Table 3. We add some sentences and phrases that include comparison of the intermediate NN to both the naïve and physics-constrained NNs in Sections 3.1, 3.2, 3.4, and 3.5. These are mostly additions and are visible as tracked changes in the revised manuscript.

**For Section 4 (Conclusions), we alter the second paragraph to reflect on the additional insight gained from including an intermediate NN. This paragraph now reads:**

Adding additional parameters based on physical information (in this case chemical reaction rates) improve predictions, as demonstrated by the intermediate NN and the physics constrained NN. Like previous work (Silva et al, 2021b) these NNs more accurately predict edge cases than the naïve NN: in our case, lower  $\Delta C$  conditions after the first hour of simulation approaching pseudo equilibrium. However, improved accuracy of the intermediate NN does not correspond to an adherence to physical laws. Both the naïve and intermediate NNs deliver solutions outside of the solution space of the reference model, including negative tendencies for purely buildup species and positive formation of species that were purely reactants. Their predictions also lead to high numbers of negative concentrations, which are nonphysical. Most importantly, material is not conserved by either the naïve NN or intermediate NN: only the physics-constrained obeyed the stoichiometric atom balance that is a fundamental property of chemical reactions. The results of this study show promise for hybrid models that combine our knowledge of physical processes with data-driven machine learning approaches, and motivates future exploration of other physically interpretable machine learning techniques that can incorporate additional prior information such as pseudo steady-state approximations.

**2) This comment concerns naming conventions.**

You chose to refer to your network as a physics constrained neural network. This name sounds similar to the one chosen by Raissi et al. (who you refer to), regarding physics informed neural networks (PINNs) which are now a widely shared name and concept in the community.

In your introduction, you clearly state that PINNs are neural networks which satisfy a given partial differential equation (PDE). Specifically, the physical information is carried via the PDE and enforced in the cost function, and the only inputs to a PINN are the variables that you differenciate with regards to in the PDE. Clearly, your network does not satisfy such a definition and you do not claim so.

Yet there are several occurences where your wording does not carry that distinction so clearly, or could make the reader confused. More preciselly, there are several occurences of the phrases "physics-informed input" or "physically informed input", such as in lines 116, 126, 130, 138 and 491. Refering to the addition of relevant bilinear terms as an input to your network in such a way is extremely confusing with regards to the widely shared definition of what a PINN represents.

Even though calling your network a phyiscs constrained neural network seems justified and different enough to me, I believe that you should rather refer to your input modification as a "reactivity-informed" input, or any other better suited name of your choice. Currently, I believe that your naming convention could generate a lot of confusion between widely different implementations.

Response: We do not want to call this "reactivity-informed" input, because this approach is more general than the example of photochemistry. We have changed this to "additional input" in all relevant cases, for example the title of section 2.2

2.2 Additional input to the neural network

**and provided an additional statement on lines 180-185 for context:**

We avoid use of physically-informed input to describe this approach, to prevent confusion with the physics-informed NN approach as introduced by Raissi et al. (2019). We do not call this approach "reactivity-informed" input, to keep the generalizability of this approach in mind. This additional input is informed by knowledge of bimolecular rate reactions: however, for other applications, additional input can take other forms. For the example of evaporation or condensation, the driving force of a concentration gradient could be supplied as additional input.

I also believe that the reference to PINNs regarding the addition of the stochiometric matrix A in line 102 is unsupported.

**Response:** We agree. Our approach is distinct from the PINNs – this phrase is removed in the updated version of the manuscript.

Please disregard comment 3, as I have since noticed that the link redirecting to the paper I mentionned was a truncated version of the link you provide.

However you might still want to edit this link, as I have noticed you provide links to both your data generation and machine learning source code in a dedicated section, and the link on line 149 refers to a previous version of the data generation source code you provide in this section.

Thank you for sharing your code.

Response: Thank you – this is fixed in line 249 with an updated link.

**Additional Changes**

To improve accessibility and clarity, and cite additional related work, we have made the following minor additions.

Line 104: We added the following sentence to address relevant related work that calculated fluxes to add physical consistency to machine learning algorithms. However, *S* values are not always standard output of such models: in some cases, the reference model can be altered to calculate and output *S* values for the ML algorithm, for example calculating subgrid fluxes to train a physically consistent NN in a climate model (Yuval et al., 2021) or using explicit Euler integration in a simplified photochemical mechanism (Sturm and Wexler, 2020). Unfortunately *S* values often cannot be readily gleaned from the reference model for training a machine learning tool, especially when more sophisticated integrators are used.

Line 400: Removed "both corresponding to observations with high changes in concentration" as it was not particularly relevant and could be confusing.

Appendix A – We have updated this paragraph to cite recent work towards ML surrogate models of aerosol microphysics (Gettelman et al., 2021; Harder et al., 2021), as well as repose the motivation: use fluxes as target properties for mass conservation. The first paragraph now reads: This framework has been demonstrated for photochemistry but can be generalized to other applications, such as change of concentrations of condensable species in a sectional aerosol model, for example MOSAIC (Zaveri et al., 2008). Recent work has been published on machine learning surrogate models for aerosol microphysics (Gettelman et al., 2021; Harder et al., 2021). Harder et al. (2021) have indicated mass conservation as a future research direction for ML surrogate models of aerosol microphysics, and have proposed regularization via a cost function or post-prediction mass fixers. Training ML algorithms on target fluxes S rather than tendencies  $\Delta C$  would allow for mass conservation to numerical precision if the tendencies are related to fluxes via an A matrix as in Sturm and Wexler (2020). Studying the system of equations modeling evaporation and condensation in a sectional model, we see that a left pseudoinverse of the corresponding A matrix can be used to obtain fluxes S from concentrations (typical model output), unlike the rank-deficient A matrix in the photochemical application focused on in this work and Sturm and Wexler (2020).

Conclusion: we move the second two sentences of the second paragraph to become the last two sentences of the first paragraph, and alter them slightly. They now read: As an example of how this framework can be implemented, we design a physics-constrained neural network surrogate model of photochemistry with input resembling bimolecular reaction rates, and a penultimate hidden layer enforcing an atom balance. The weights for the penultimate layer are hard stoichiometric constraints and can be obtained via the approach in Sturm and Wexler (2020) relating tendencies of molecular species to atom fluxes between them.

**Code and data availability**

This section now contains an updated Zenodo doi for the Python code and downloadable neural networks.

---

## Author Response (AR2)

Author response to referee technical corrections on
**Conservation laws in a neural network architecture: Enforcing the atom balance of a Julia-based photochemical model (v0.2.0)**
P. Obin Sturm and Anthony S. Wexler
29 March 2022

We would like to thank the referees and the editor for their time. Below are point-by-point responses to technical corrections required for acceptance of the final manuscript. Points from referee reports are in *italic*, our responses are in **bold**, and updates to the manuscript are shown in blue.

Report #1
Submitted on 25 Mar 2022
Anonymous referee #2
*For final publication, the manuscript should be accepted subject to technical corrections.*

*The authors have addressed my previous comments/questions, however, I am still puzzled with the answer to the following:*
*Line 115: ... several key differences:*
*1. Our entire output vector represents a coupled system where all elements are subject to the constraints. This differs from the approach in Beucler et al. (2021). which constrains a chosen subset of the output and allows some output to be unconstrained.*

*My point was: The approach in Beucler et al allows to constrain any subset of the output, and the subset can also be the whole output. If you restrict to only allowing to constrain all the output, I would say it is just a particular or restricted case, right? If that is the case, then why is the difference a key difference in positive sense?*
**Response: Our approach is not a subcase of the Beucler et al. (2021) approach, as our formulation using the A matrix does not require relating inputs to the outputs, but rather balances tendencies in the output vector. This core difference can be found by comparing Eqns. 1 and 2 and corresponding discussion in this manuscript to Eq. 1 in the cited Beucler et al. (2021) publication. This difference is also visually illustrated by comparing Figure 1 in this manuscript to Figure 2 in the Beucler et al. (2021) publication, where inputs are fed to the constraints layers. Due to this fundamental difference, we do not want to make the argument that this is a particular or restricted case. We do note, however, that our approach as developed here operates on all output and is therefore more restrictive than the Beucler et al. (2021) approach (though our approach could conceivably be generalized to only enforce a flux continuity constraint on a portion of the output, such a generalization is outside of the scope of our example). Thanks for pointing out that this point could be made clearer; we have refined this point to now read:**

1. Our entire output vector represents a coupled system where all elements are subject to the constraints. This formulation is more restrictive than the approach in Beucler et al. (2021), which constrains a chosen subset of the output and allows some output to be unconstrained.

Report #2
Submitted on 28 Mar 2022
Anonymous referee #1
*For final publication, the manuscript should be accepted subject to technical corrections.*

*The paper is in good shape. I only have a few technical corrections*

*l 60: "these models" should read "air quality and climate models"*
**Response: Done!  Note that this is line 48 now – the previous manuscript had a discrepancy where the count was skipped for 12 lines (somewhere in between 30 and 45).**

*eq. 3 brackets are in subscript*
**Response: We have checked to make sure that the parentheses in equation 3 are not in the subscript.**

*Figure captions: reference to intermediate model is missing in Figure 2, 3, B1, B2.*
**Response: Thank you for catching this.  All four captions are now updated to begin with:**
Scatter plots of target values to predicted values, for the naïve NN (orange, top row), the intermediate NN (purple, middle row) and the physics-constrained NN (green, bottom row)…

*l 555: contrast, not contract*
**Response: We have replaced this word in section 3.5 accordingly.**

*l 644: I would call this a cloud microphysical model rather than an aerosol microphysical model*
**Response: We have updated this sentence, which now reads:**
Recent work has been published on machine learning surrogate models for cloud microphysics with resolved size bins (Gettelman et al., 2021) and aerosol microphysics using a modal approach (Harder et al., 2021).